# STRUCTURED ATTENTION MATTERS TO MULTIMODAL LLMS IN DOCUMENT UNDERSTANDING

## ABSTRACT

Document understanding remains a significant challenge for multimodal large language models (MLLMs). While previous research has primarily focused on locating evidence pages through precise multimodal queries, our work investigates a fundamental yet overlooked aspect: how input format influences document comprehension performance. Through systematic analysis, we discover that plain multi-element text extracted from PDFs often impairs rather than improves MLLMs' performance, a counterintuitive finding that we attribute to attention dispersion and loss of structure. To further substantiate our hypothesis, we propose using the LaTeX paradigm as a tool for encoding document elements, maintaining the hierarchical organization and spatial relationships critical for comprehension. Our attention analysis reveals that structured multi-element text induces structured attention patterns in both textual and visual content, directing models to focus on semantically meaningful regions while reducing attention waste. Specifically, we found that the structured text significantly enhance MLLMs' document question-answering performance across diverse document types without requiring architectural modifications or additional training.

## 1 INTRODUCTION

Document understanding is a common task in daily life. With the rapid development of multimodal large language models (MLLMs), capable general-purpose systems are increasingly expected to comprehend documents effectively (Zhu et al., 2023; Liu et al., 2024a; 2023; Wang et al., 2024a; Bai et al., 2023; Team, 2025; Anthropic, 2023; Dong et al., 2024; Chen et al., 2024b; Li et al., 2023; Wang et al., 2024b). The difficulty in document question answering arises from the diverse nature of questions and the variety of information required, which evidence elements include text blocks, charts, diagrams, and figures, often requiring integration of multiple information sources. Document understanding through MLLMs presents three key challenges: (1) the information diversity challenge of processing heterogeneous elements, (2) the context integration challenge of synthesizing scattered information, and (3) the structural relationship challenge of understanding how elements relate spatially and logically—relationships intuitive for humans but difficult for machines (Han et al., 2025).

Previous research has addressed these challenges primarily by extending context windows to accommodate more content or by developing specialized architectures for extracting multi-granularity information (Ding et al., 2022; Hu et al., 2024; Ye et al., 2023; Liu et al., 2024b; Park et al., 2024; Tito et al., 2023). With general-purpose MLLMs, the trend has shifted toward retrieval-augmented generation (RAG), which locates relevant evidence and feeds it into models as images, text, or both (Mishra et al., 2019; Suri et al., 2024; Park et al., 2024). Despite these advances, a critical question remains unexplored: how does the format of input information, rather than merely its content—influence document understanding? Current approaches typically extract text but discard critical structural information (Khattab & Zaharia, 2020; Faysse et al., 2024). This leads to a counterintuitive phenomenon we observed across multiple datasets: unstructured plain multi-element text extracted from PDFs often degrades rather than enhances MLLM performance compared to using images alone (Cho et al., 2024; Deng et al., 2024; Han et al., 2025; Zhang et al., 2024a). This observation motivated us to investigate the relationship between input structure and model comprehension. We discovered that information structure fundamentally shapes how MLLMs allocate

Figure 1: Comparison of Different Approaches for DocQA: Previous research methods focused on using RAG to precisely locate the evidence and then directly input the evidence into general-purpose MLLMs, or on designing task-specific models that focus on multi-granularity extraction of image information and expanding the context window. We propose a novel structure-preserving approach based on the LATEX paradigm to explore the impact of input format on general models' responses to DocQA and investigate potential causes through attention analysis.

attention across document elements. With unstructured text, models exhibit scattered attention patterns, wasting computational resources and struggling to identify relationships between elements.

Our study includes two main steps. First, we propose an approach to preserve the structure of plain multi-element text extracted from PDFs. We input the evidence plain text and images into an MLLM and prompt it to generate structured text. This step aims to maintain the structure of diagrams, tables, and texts. The layout information can be preserved in the form of text, which is helpful during the answer generation process. We input the images along with the LATEX paradigm to generate answers, exploring the importance of document layout and the structure of sub-elements on the evidence page. We find that the accuracy of MLLMs in answering questions is significantly improved. Second, we analyze the attention distribution and transformation with different inputs, comparing the cases of using images alone and images combined with structured text as input. We find that attention is more focused when constrained by the structured input.

Overall, our findings show that the ability of general-purpose models in document understanding can be improved simply by changing the input format. Furthermore, the attention transformation between different inputs demonstrates that the structured attention brought by structured input is key to improving the ability of MLLMs to answer DocQAs. This indicates that there are other aspects worth exploring beyond focusing solely on enhancing the effectiveness of Retrieval Augmented Generation (RAG).

## 2 RELATED WORK

We discuss two lines of related work: MLLMs in DocQA tasks and methods used in DocQA tasks.

**MLLMs in DocQA Tasks.** Recent Document Visual Question Answering (DocVQA) models focus on expanding context window sizes, improving fine-grained visual comprehension, and enhancing layout analysis (Hu et al., 2024; Liu et al., 2024b; Tworkowski et al., 2023). These systems often boost performance through architectural enhancements (e.g., added modules) or multi-stage training pipelines with diverse data inputs (Park et al., 2024; Tito et al., 2023; Chen et al., 2023). While general multimodal models like Gemini (Team et al., 2024) and Qwen-VL (Bai et al., 2023) show improved visual processing and context handling, they remain constrained by input length limitations and multi-page processing capabilities. Their reliance on image-derived representations also hinders precise localization of detailed elements, while layout extraction from visual data risks diverting the model's focus, reducing answer accuracy (Han et al., 2025).

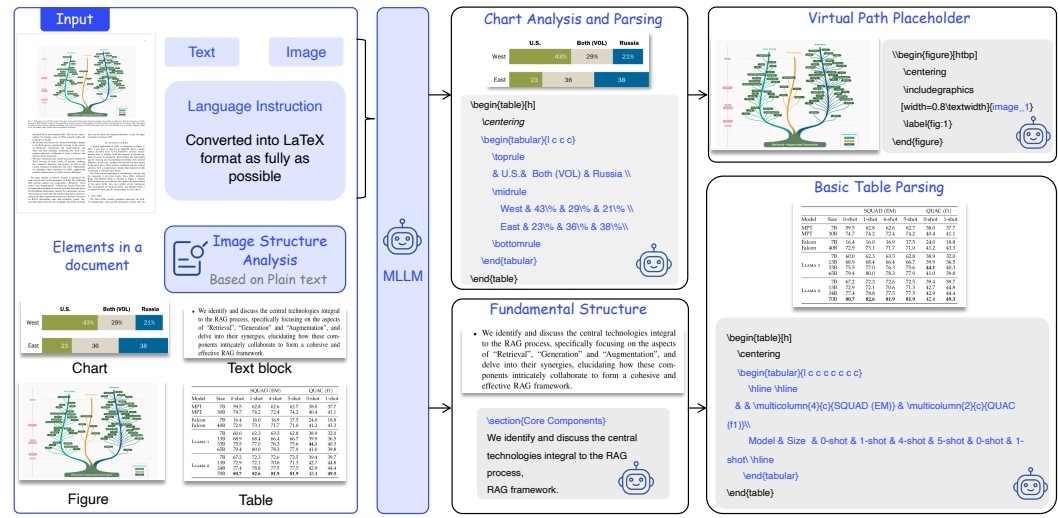

Figure 2: Structured text is generated using LATEX. We prompt the MLLM to capture the layout of the given images as accurately as possible, producing blocks that include text, charts, and tables. Figures that cannot be parsed into concrete content are represented using virtual paths and the LA-TEX paradigm. This approach is simple and only requires API-level access with instruction-level control.

**Methods in DocQA Tasks.** Retrieval augmentation has become pivotal for adapting general multimodal models to document question answering, addressing input constraints and information overload by retrieving fine-grained evidence through multimodal or unimodal queries (Lewis et al., 2020; Gao et al., 2023; Chen et al., 2024a; Cho et al., 2024). While current methods enhance accuracy through optimized retrieval strategies (Deng et al., 2024; Lu et al., 2024), input formatting remains challenging: image-only inputs hinder textual comprehension due to incomplete integration, whereas plain multi-element text extracted from PDFs neglect visual elements critical for image-dependent questions. Multimodal inputs combining text and images risk overwhelming models with redundant or conflicting data, leading to attention dispersion. To maximize performance, refining input strategies—such as modality prioritization, structured evidence fusion, or dynamic filtering—is essential for balancing information richness and focused reasoning in retrieved evidence.

## 3 METHODOLOGY

Understanding documents through multimodal large language models (MLLMs) represents a complex challenge at the intersection of vision and language processing. Other research mainly investigates the importance of locating precise evidence pages that are directly input into MLLMs, while often ignoring how the input representation influences the performance in document question answering.

In this section, we propose using the LATEX paradigm as a tool for retaining document structure. This is a novel structure-preserving approach that encodes multiple elements in documents, which are then transferred into structured multi-element text. Images combined with structured text significantly enhance the MLLMs' document understanding capabilities without requiring architectural modifications or additional training. We analyze the attention transformation and question answering ability under different inputs, revealing that structured multi-element text induces structured attention patterns on both textual and visual content, collectively enabling the comprehension of MLLMs.

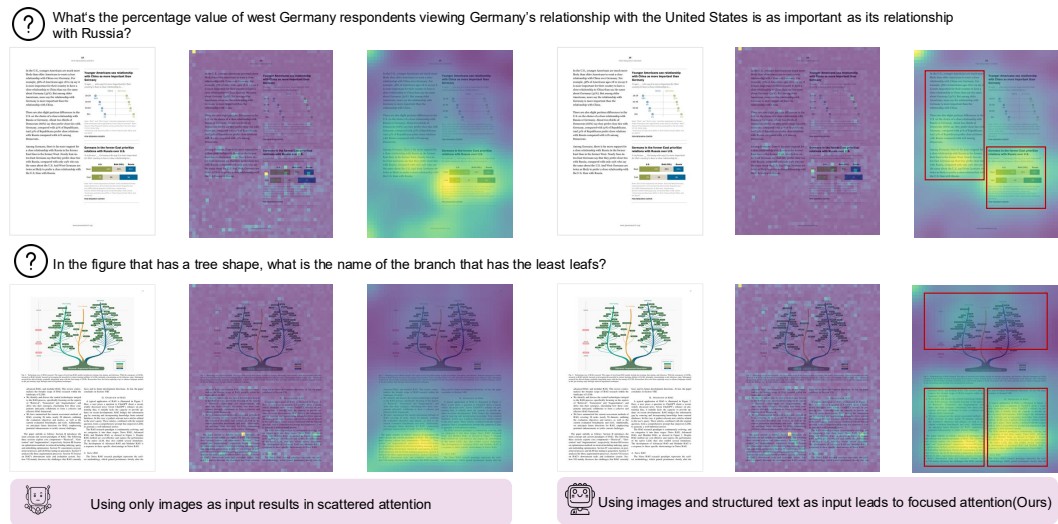

Figure 3: An example of attention transformation under two conditions: images alone versus images combined with structured text. The MLLMs tend to focus on text blocks, figures, and charts, rather than the overall layout.

## 3.1 INPUT REPRESENTATION IN DOCUMENT UNDERSTANDING

Document understanding requires models to interpret diverse information types—text blocks, tables, figures, charts, and their interrelationships. Through systematic analysis of MLLM performance in document question answering (DocQA) tasks, we identified a critical paradox.

**Observation 1.** *Providing unstructured multi-element text alongside document images often degrades MLLM performance across benchmark datasets, despite increasing the total information available to the model.*

This counterintuitive finding contradicts the common assumption in retrieval-augmented generation that more textual context improves performance. To investigate this phenomenon, we conducted a series of controlled experiments comparing MLLMs' attention patterns when processing: (1) document images alone, (2) images with unstructured plain multi-element text, and (3) images with our proposed structured text representation.

## 3.2 PLAIN MULTI-ELEMENT TEXT WITH STRUCTURE

Based on observation 1, we propose an approach to preserve the structure of the plain multi-element text extracted from PDFs. As shown in Figure 2, we use LATEX to encapsulate the plain text. It is easy for MLLMs to understand LATEX code because it usually follows a fixed and easy-to-understand paradigm. The LATEX code constrains the content in titles, tables, and figures, providing efficient and additional references when MLLMs answer DocQAs with images. We input the image and the corresponding plain text into the MLLM to obtain the structured text. We prompt the MLLM to preserve the structure and text related to charts in the image as much as possible. If a figure in the image cannot be converted into LATEX content, we instruct the MLLM to use a virtual path to represent the figure while keeping the structure intact. An example is provided in Figure 2.

We use the proposed method to obtain structured text and compare the performance of MLLMs under three cases: images only, images with plain text, and images with structured text. The results confirm the hypothesis in Section 3.1, leading to the following observation:

**Observation 2.** *Structured text and images work together to improve MLLMs' performance in answering DocQAs.*

Observation 2 shows that the performance of MLLMs are improved significantly simply by preserving the structure of the input text, without the need to provide additional fine-grained positional

information to help answer the questions. This observation proves the importance of structure, which means that merely informing the MLLMs of the structure of document helps them gain a better overview and answer questions correctly.

### 3.3 ATTENTION TRANSFORMATION WITH STRUCTURED TEXT

Given observation 2, this subsection seeks to uncover the underlying reasons why structured text matters by analyzing the attention distribution when MLLMs answer questions using only images versus using structured text as references. Figure 4 shows that in the structured case, the attention scores are less sensitive to the boundaries of the image and more concentrated on the main body, indicating that the MLLMs know where to focus under the constraints of structured text. Based on these findings and experimental results, we have the following observation:

**Observation 3.** *Structured text brings structured attention to both texts and images, which directly improves the abilities of MLLMs. This shows that structured attention is the key to helping MLLMs answer DocQAs.*

Observation 3 demonstrates the necessity of structure in document understanding. As shown in the example in Figure 3, structured text constrains MLLMs and reduces irrelevant attention when focusing on images. This helps MLLMs recover distracted attention and focus on images and relevant text to answer the given question.

## 4 EXPERIMENTS

We conduct evaluations on four document understanding benchmarks covering multiple scenarios to provide solid evidence supporting the observations presented in this paper.

### 4.1 DATASETS AND MODELS

**Implementation Details.** We apply the document preprocessing method from Han et al. (2025) to obtain plain multi-element text using a combination of Optical Character Recognition (OCR) and PDF parsing techniques, together with visual content obtained by transforming pages in long documents into images. Specifically, OCR is employed to recognize text within image-based PDFs, while PDF parsing extracts text directly from digitally encoded text within the PDF. This dual approach ensures robust text extraction across various document formats and structures. We then locate the evidence pages as images, from which we extract the corresponding plain text. We subsequently apply the approach presented in Section 3.2 to obtain the structured text. All experiments are conducted on 4 NVIDIA A100 GPUs.

**Models.** Our evaluation uses the following four multimodal LLMs: QWEN2-VL-7B-INSTRUCT, QWEN2.5-VL-7B-INSTRUCT, LLAVA-V1.6-MISTRAL-7B and PHI-3.5-VISION-INSTRUCT. These models accept both images and text as input and generate answers and analyses for the given questions.

**Datasets.** The benchmarks include MMLongBench (Ma et al., 2024), LongDocUrl (Deng et al., 2024), PaperTab (Hui et al., 2024), and FetaTab (Hui et al., 2024). These evaluation datasets cover a variety of scenarios, including open- and closed-domain, textual and visual, as well as long and short documents, ensuring fairness and completeness in the evaluation.

**Metrics.** For all benchmarks, we leverage LLAMA-3.1-8B-INSTRUCT as the evaluation model to assess the consistency between the model's output and the reference answer producing a binary decision (correct / incorrect). We report the average accuracy for each benchmark.

### 4.2 IMPACT OF PLAIN MULTI-ELEMENT TEXT

We first compare the performance of MLLMs when using images alone versus using images with the help of plain multi-element text. According to Table 1, the results show that plain text impairs MLLMs' performance on MMLongBench, which contains multi-element evidence images including charts, tables, blocks of text, and figures. The additional text information decreases the accuracy of

Table 1: Our proposed structure-preserving method effectively retains the document structure and significantly improves the accuracy of DocQA responses. For the open-domain dataset, we followed the retrieval method used in Han et al. (2025), using the top-1 retrieval result as input. On the closed-domain dataset, we used the evidence pages of the dataset as input to test the importance of document structure.

| Model | MMLongBench | LongDocUrl | PaperTab | FetaTab | Avg |
|---|---|---|---|---|---|
| **Prior Multimodal LLMs** | | | | | |
| QWEN2-VL-7B-INSTRUCT w/ image (Wang et al., 2024a) | 0.292 | 0.195 | 0.160 | 0.441 | 0.272 |
| QWEN2-VL-7B-INSTRUCT w/ image+text (Han et al., 2025) | 0.287 | 0.219 | 0.188 | 0.507 | 0.300 |
| QWEN2.5-VL-7B-INSTRUCT w/ image (Team, 2025) | 0.389 | 0.197 | 0.163 | 0.508 | 0.314 |
| QWEN2.5-VL-7B-INSTRUCT w/ image+text (Han et al., 2025) | 0.375 | 0.399 | 0.229 | 0.557 | 0.427 |
| LLAVA-V1.6-MISTRAL-7B w/ image (Liu et al., 2024a) | 0.131 | 0.126 | 0.051 | 0.154 | 0.116 |
| LLAVA-V1.6-MISTRAL-7B w/ image+text (Han et al., 2025) | 0.183 | 0.159 | 0.127 | 0.406 | 0.219 |
| PHI-3.5-VISION-INSTRUCT w/ image (Abdin et al., 2024) | 0.189 | 0.131 | 0.077 | 0.245 | 0.161 |
| PHI-3.5-VISION-INSTRUCT w/ image+text (Han et al., 2025) | 0.226 | 0.213 | 0.160 | 0.443 | 0.261 |
| **Prior Multimodal LLMs + Structured text (Ours)** | | | | | |
| QWEN2-VL-7B-INSTRUCT w/ image | **0.306** | **0.229** | **0.209** | **0.509** | **0.313** |
| + **Structured Text** | +4.8 % | +17.4% | +30.6% | +15.4 % | |
| QWEN2.5-VL-7B-INSTRUCT w/ image | **0.435** | 0.221 | **0.252** | **0.575** | 0.371 |
| + **Structured Text** | +11.8% | +12.2% | +54.6% | +13.2% | |
| LLAVA-V1.6-MISTRAL-7B w/ image | **0.224** | 0.153 | 0.122 | 0.388 | **0.222** |
| + **Structured Text** | +71.0% | +21.4% | +139.2% | +151.9 % | |
| PHI-3.5-VISION-INSTRUCT w/ image | **0.284** | 0.211 | **0.224** | 0.429 | **0.287** |
| + **Structured Text** | +50.3 % | +61.1% | +190.9% | +75.1% | |

the large model on this dataset from 0.389 to 0.370. The accuracy of MLLMs on other datasets might increase, but this can also impair their ability in specific cases. Providing unstructured plain text alongside document images sometimes degrades MLLM performance across benchmark datasets, despite increasing the total information available to the model. These results and the case shown in Figure 5 clearly and convincingly support observation 1, showing that repeated and redundant text information does not help large models answer questions about long documents.

We hypothesize that the underlying cause lies in the loss of structural information during the conversion to plain text. When relevant elements in document images—such as text blocks, charts, and figures—are transformed into plain text, their original structural constraints are discarded. Consequently, MLLMs face difficulties in accurately interpreting both the individual elements and the relationships among them, which in turn results in information confusion and diminishes their effectiveness in answering document-centric questions.

### 4.3 PERFORMANCE IMPROVEMENTS WITH STRUCTURED TEXT

We consider the impact of structure on the process of MLLMs understanding documents. We compare using structured text as a reference with using only images for understanding. On MMLong-Bench, adding structured text information increases the accuracy of QWEN2.5-VL-7B-INSTRUCT from 0.389 to 0.435. This 10% performance improvement fully demonstrates the importance of structure in long document reading. The structure of different elements in the evidence images is preserved as much as possible due to the instructions given to the MLLM. The structured text surprisingly enhances the ability to answer questions. On MMlongBench, every models' performance are improved because almost all evidence pages in this dataset contain structured tables and graphs, which can be fully interpreted by the LATEX paradigm combined with plain text. Additionally, the accuracy of QWEN2.5-VL-7B-INSTRUCT is significantly higher compared to QWEN2-VL-7B-INSTRUCT, demonstrating that QWEN2.5-VL-7B-INSTRUCT has a stronger ability to understand LATEX and make better use of structure.

The corresponding data further observation 2. By employing our proposed novel structure-preserving method, which leverages an intuitive LaTeX paradigm, we are able to retain both the internal structure of individual elements within document images and the relationships among these elements. Without requiring additional training or modifications to the model architecture, this approach enhances the ability of MLLMs to comprehend documents, thereby demonstrating the critical importance of structural information.

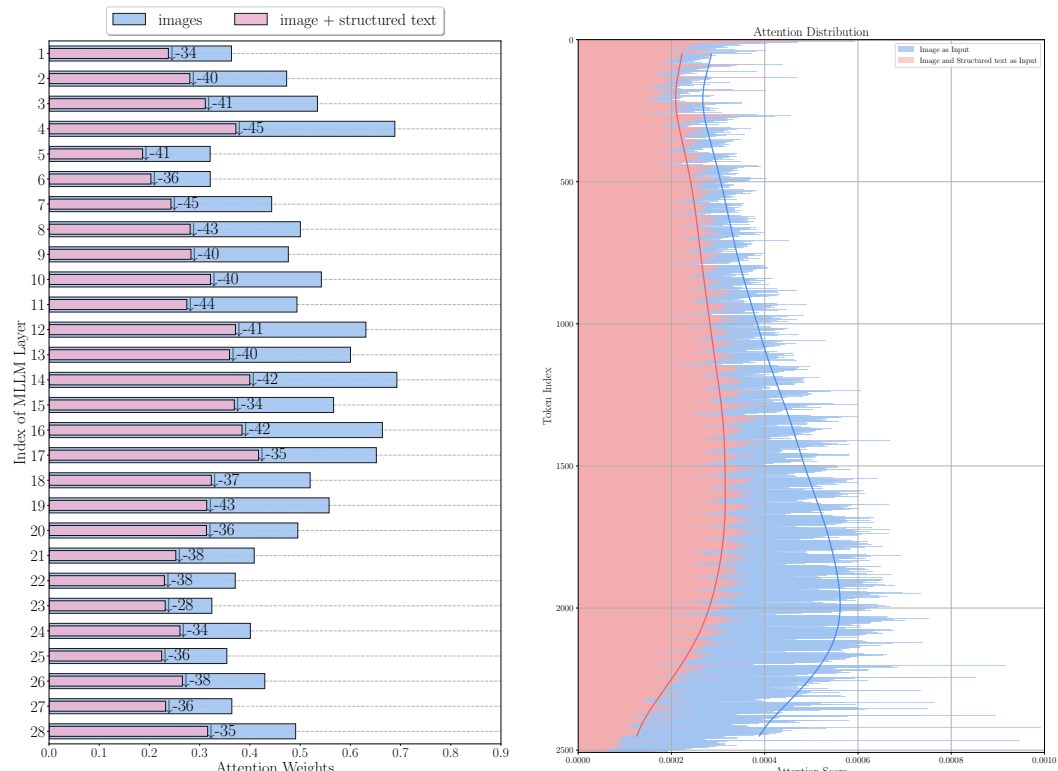

Figure 4: (a) Comparison of attention weights across different layers of MLLMs for images alone versus images combined with structured text. (b) MLLMs are less sensitive to image border tokens when constrained by structured text. The attention distribution shows that attention scores at the borders are lower in the presence of structured text.

## 4.4 ATTENTION ANALYSIS

We follow Zhang et al. (2024b) to conduct attention analysis in order to understand why structured text matters to MLLMs. We analyze attention from two perspectives: attention distribution and attention transformation across different cases. We present the following findings to support Observation 3.

**Attention Distribution.** We consider 370 samples with the same number of image tokens to demonstrate the distribution of attention transformation with or without structured text as references when MLLMs answer questions based on evidence images. We use QWEN2.5-VL-7B-INSTRUCT as the main model to generate answers on MMLongBench. In the case where MLLMs rely only on images, the attention distribution shows that the model is sensitive to boundary tokens of images and exhibits an uneven distribution of attention. These results indicate that MLLMs treat almost every image token equally and have no clear focus on specific regions of the original images. In contrast, MLLMs are constrained when using structured text as references. MLLMs learn to focus on useful image tokens located in figures, blocks of text, or tables. This transformation of attention distribution highlights the importance of structured text.

Figure 4 illustrate the attention distribution of QWEN2.5-VL-7B-INSTRUCT under different inputs, where the former shows the overall variation across 28 layers and the latter presents the attention changes within a single layer. We observe that the input of structured text induces more focused attention and reduces attention loss at the page boundaries. This indicates that structured text contributes to a dual form of structured attention—spanning both images and text—which becomes a primary factor in enhancing the document comprehension ability of MLLMs.

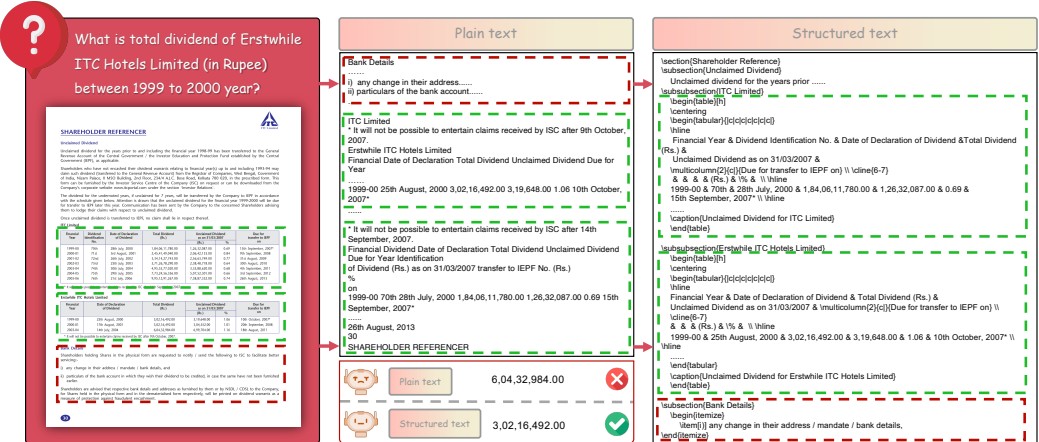

Figure 5: A comparison of generated answers from MLLMs using plain multi-element text versus structured text shows that structured text improves MLLM performance on DocQA tasks. The LA-TEX paradigm helps preserve the image's structure, aiding the model in locating evidence relevant to the question.

**Case Study.** We perform a case study to better understand attention transformation with the help of structured text. Figure 5 illustrates an example. The question requires MLLMs to find the percentage of West Germany respondents who view Germany's relationship with the United States as equally important as its relationship with Russia, based on the chart and corresponding text. The attention map in Figure 3 shows that attention is distributed everywhere without the control of structured attention, even in the blank areas of the given image. With the constraint of structured attention, attention focuses on blocks of text, charts, and so on. We conclude that structured text helps MLLMs reduce attention loss, guides them where to look, and increases the probability of finding fine-grained evidence regions. All these constraints enable MLLMs to better understand documents and answer DocQAs more effectively.

## 4.5 ABLATION STUDIES

We almost obtain the best results on almost every dataset and model by using structured text and images as input. We further conduct experiments on the following cases: structured text as input, plain text as input, and the LATEX format acting as a placeholder without specific text information. Based on these cases, we aim to demonstrate the necessity of combining structured text and

Table 2: Performance comparison on different datasets. The ablation results show the importance of combining structured text and image. When structured text is combined with an image, it results in structured attention, which eventually helps MLLMs answer questions.

| Model | MMLongBench | LongDocUrl | Avg |
|---|---|---|---|
| QWEN2-VL-7B-INSTRUCT w/ text | **0.291** | **0.175** | **0.233** |
| QWEN2-VL-7B-INSTRUCT w/ structured text | 0.212 | 0.163 | 0.188 |
| QWEN2.5-VL-7B-INSTRUCT w/ text | **0.318** | **0.188** | **0.253** |
| QWEN2.5-VL-7B-INSTRUCT w/ structured text | 0.267 | 0.155 | 0.211 |
| LLAVA-V1.6-MISTRAL-7B w/ text | **0.287** | **0.162** | **0.225** |
| LLAVA-V1.6-MISTRAL-7B w/ structured text | 0.246 | 0.151 | 0.199 |
| PHI-3.5-VISION-INSTRUCT w/ text | **0.299** | **0.185** | **0.242** |
| PHI-3.5-VISION-INSTRUCT w/ structured text | 0.244 | 0.167 | 0.206 |

Table 3: Performance comparison across different evidence sources on MMLongBench reveals how various document elements demonstrate the differing effects of structured input on document understanding.

| Model | Chart | Table | Pure-Text | Generalized-text | Figure | Avg |
|---|---|---|---|---|---|---|
| **Prior Multimodal LLMs** | | | | | | |
| QWEN2-VL-7B-INSTRUCT w/ image | 0.278 | 0.304 | 0.311 | 0.311 | 0.318 | 0.304 |
| QWEN2-VL-7B-INSTRUCT w/ image+text | 0.244 | 0.300 | 0.358 | 0.294 | 0.301 | 0.299 |
| QWEN2.5-VL-7B-INSTRUCT w/ image | 0.392 | 0.350 | 0.427 | 0.387 | 0.418 | 0.395 |
| QWEN2.5-VL-7B-INSTRUCT w/ image+text | 0.375 | 0.367 | 0.374 | 0.420 | 0.388 | 0.285 |
| LLAVA-v1.6-MISTRAL-7B w/ image | 0.120 | 0.060 | 0.116 | 0.126 | 0.134 | 0.111 |
| LLAVA-v1.6-MISTRAL-7B w/ image+text | 0.114 | 0.106 | 0.195 | 0.151 | 0.154 | 0.144 |
| PHI-3.5-VISION-INSTRUCT w/ image | 0.176 | 0.147 | 0.185 | 0.160 | 0.244 | 0.182 |
| PHI-3.5-VISION-INSTRUCT w/ image+text | 0.205 | 0.203 | 0.291 | 0.261 | 0.224 | 0.237 |
| **Prior Multimodal LLMs + Structured text (Ours)** | | | | | | |
| **QWEN2-VL-7B-INSTRUCT w/ structured text** | **0.307** | 0.267 | 0.348 | 0.261 | 0.288 | 0.294 |
| **QWEN2.5-VL-7B-INSTRUCT w/ structured text** | 0.364 | 0.364 | 0.374 | 0.328 | 0.385 | 0.363 |
| **LLAVA-v1.6-MISTRAL-7B w/ structured text** | **0.159** | **0.129** | **0.205** | **0.160** | **0.161** | **0.163** |
| **PHI-3.5-VISION-INSTRUCT w/ structured text** | **0.261** | **0.263** | **0.315** | 0.227 | 0.231 | **0.259** |

images. Table 2 shows that cases relying only on plain text or only on structured text result in poor performance of MLLMs. Structured text is ineffective when not input alongside images.

We conducted ablation experiments categorized by different evidence elements from MMLong-Bench. The results in Table 3 indicate that for questions in documents that necessitate answers from charts or tables, using structured input provides a more pronounced performance gain than for other types of document elements. This observation further underscores the critical role of structured input.In the cases needed images to answer questions, the accuracy is lower. This is understandable because the lack of images causes information loss in cases where MLLMs need figures from the evidence images to answer questions. The accuracy with structured text ans images is better than with plain text and images, further highlighting the importance of structure. This experimental result convincingly demonstrates the importance of structure. We conclude that the combination of structured text and images is key to improving the ability of MLLMs in long document understanding, supporting Observation 3 from an ablation perspective.

## 5 CONCLUSION

Preserving the structure of the input to improve the ability of general-purpose multimodal MLLMs is essential to comprehend the underlying patterns and key factors in document understanding. This work makes a step in this line. We propose to use LATEX paradigm to keep the structure of plain text, the images combined with structured text efficiently improve the accuracy of answering questions in document understanding. Furthermore, we analyze the attention transformation of different kinds of input. This study find the structured attention is the key to make MLLMs understand better in answering DocQAs after comparing other inputs. Future work includes proposing novel and efficient structure extraction or attention control method to effectively unlock the ability of general-purpose MLLMs in document understanding.

## 6 ETHICS STATEMENT

This research adhered to ethical guidelines, ensuring proper consent, data privacy, and confidentiality. No conflicts of interest are reported, and the findings are presented transparently.

## 7 REPRODUCIBILITY STATEMENT

All relevant code, data, and materials are publicly available at `https://anonymous.4open.science/r/structured-attention_anonymous-8994/`, with documentation provided for reproducibility. Any limitations are clearly stated.

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

# A   APPENDIX

## A.1   USE OF LARGE LANGUAGE MODELS (LLMs)

In the preparation of this manuscript, Large Language Models (LLMs, e.g., ChatGPT) were used solely as a language editing tool to improve readability, grammar, and style. They were not involved in research ideation, experimental design, data analysis, or the formulation of scientific conclusions. The authors take full responsibility for all scientific content and claims presented in this paper.

