# OpenReview forum: "Structured Attention Matters to Multimodal LLMs in Document Understanding"
_ICLR.cc/2026/Conference — Submitted to ICLR 2026_

### Official Review · Reviewer_8ERK · 2025-10-28

**Soundness:** 2
**Presentation:** 3
**Contribution:** 1
**Rating:** 2
**Confidence:** 4

**Summary:**

This work mainly shows unstructured text extracted from PDFs can harm MLLMs' document understanding by dispersing attention and losing layout information while representing document elements in LaTeX can preserve structural relationships and focused the attention scores on correct elements. Experiments on multiple benchmarks reveal that structured inputs improve DocQA accuracy without retraining, guiding models toward semantically relevant regions and enhancing multimodal reasoning efficiency.

**Strengths:**

1. The paper provides a thorough validation of the conclusion that structured input leads to better document understanding than either pure image or unstructured text inputs.
2. The paper is well-written, with a clear organization. The use of stepwise “observations” in the methodology section makes the reasoning process intuitive and easy to follow.

**Weaknesses:**

1. The main conclusion, while empirically supported, appears too simple to sustain a top-tier conference contribution.
2. The control of input variables is insufficiently rigorous, differences in image resolution or text formatting can cause large variations in token counts, which may affect VQA performance and should be accounted for in analysis.
3. The attention analysis is relatively shallow. Additionally, in Figure 4, the right-side attention scores are not normalized, and the left-side comparison does not clearly convey its intended implication.

**Questions:**

See weaknesses.

---

### Official Review · Reviewer_DUqb · 2025-10-31

**Soundness:** 1
**Presentation:** 2
**Contribution:** 1
**Rating:** 2
**Confidence:** 4

**Summary:**

The paper investigates how input format affects Multimodal Large Language Models (MLLMs) in document understanding tasks. The authors observe that plain text extracted from PDFs often degrades model performance. To address this, they propose using LaTeX-formatted structured text as input alongside images, aiming to preserve document layout and element relationships. They conduct experiments on multiple benchmarks and models (7B scale), showing that structured input improves performance and leads to more focused attention patterns. The authors claim this is a novel, training-free method to enhance document understanding without modifying model architecture.

**Strengths:**

The work is easy to follow, the datasets are standard, and the attention visualisations give a concrete illustration of why scattered tokens mislead the model. It is also useful to be reminded that “more text” is not automatically better if the layout signal is lost.

**Weaknesses:**

1. **Lack of Novelty in Observations:**
The central observation — that plain text extracted from PDFs can hurt performance — is not surprising and has been implicitly or explicitly noted in prior work. The degradation is largely due to loss of layout and structural context, which is well-known in OCR and document AI communities. Presenting this as a novel finding overstates the contribution.


2. **Methodological Depth: More Like Prompt Engineering:**
The proposed method — using LaTeX to represent document structure — is essentially a prompt engineering trick. While effective, it lacks technical novelty or algorithmic innovation. There is no learning, optimization, or generalization mechanism involved.


3. **Limited Model Scale and Generalization:**
All experiments are conducted on 7B-scale open-source models. There is no evaluation on larger models (e.g., 30B, 70B+) or proprietary models (e.g., GPT-5, Claude-4, Gemini-2.5)
These models are more robust to input format variations. The claimed benefits may not generalize to stronger models.


4. **Practical Applicability and Assumptions:**
The method assumes access to LaTeX-formatted representations of documents, which is rare in real-world applications. Most documents are scanned PDFs, PowerPoints, or HTML pages without LaTeX source. The current pipeline relies on MLLMs to generate LaTeX from images, which is error-prone and not scalable.


5. **Missing Baselines and Ablations:**
No comparison with other structured formats (e.g., HTML, XML, Markdown).

**Questions:**

Please see weaknesses.

---

### Official Review · Reviewer_bU5J · 2025-10-31

**Soundness:** 2
**Presentation:** 3
**Contribution:** 1
**Rating:** 2
**Confidence:** 4

**Summary:**

The paper propose to convert evidence pages into LaTeX “structured text” and feeding that alongside page images to improves MLLM document QA. The method is simple: prompt an MLLM to transform plain text and images into LaTeX text (tables, figures with paths, sections), then answer questions with images and this LaTeX structured text. The authors report accuracy gains on four DocQA benchmarks using several 7B-scale MLLMs compared with plain text settings and provide attention visualizations to claim “structured attention” explains the gains.

**Strengths:**

**1.** Clear, easy-to-reproduce idea (formatting as LaTeX) rather than architectural changes.

**2.** Comprehensive evaluations: including multiple datasets and models evaluation (MMLongBench, LongDocURL, PaperTab, FetaTab; Qwen2-VL/2.5-VL, LLaVA-1.6-Mistral, Phi-3.5-Vision).

**3.** Consistent experimental framing comparing images only with images+plain text and with images+structured text. This paper also includes ablations and attention visualization and analyses to support the idea.

**Weaknesses:**

**1.**  Incremental novelty: converting extracted content to a markup that preserves hierarchy/layout is not new in document understanding; many systems preserve structure via HTML/Markdown/layout tokens. The specific choice of LaTeX as the markup is a thin tweak, and the paper offers little principled justification for LaTeX compared with simpler structured formats. The “Observations 1–3” are mostly empirical restatements.

**2.** Method underspecified
It’s unclear which MLLM generates the structured text, whether it is the same as (or stronger than) the QA model, and how prompts were chosen. If a stronger or differently-trained model creates LaTeX summaries, the downstream gains might is not fair. The paper does not include any the generator model of structured data or settings.

**3.** Efficiency missing: the approach appears to expand token counts substantially, but there is no accounting of latency, context budget, or cost relative to baselines. Gains without efficiency reporting are hard to assess for practicality.

**4.** Weak baselines: stronger structured-evidence baselines (HTML/Markdown with bounding-box, existing doc-structure tools, etc.) are missing.

**5.** Attention analysis is over-interpreted: “structured attention is key” to a conclusion without solid evidence beyond visuals. with no rigorous causal link established.

**6.** Limited generality. In Table 3, with structured Latex data, the performance on several subsets (Generalized-text and Figure) is overall degraded. The proposed DocQA with structured Latex data does not show consistent gain for the general purpose.

**Questions:**

Check Weaknesses 1-6.

---

### Official Review · Reviewer_a6qk · 2025-11-26

**Soundness:** 2
**Presentation:** 3
**Contribution:** 2
**Rating:** 4
**Confidence:** 3

**Summary:**

This paper focuses on the challenges of multimodal large language models in document understanding, and finds that unstructured multi-element text extracted from PDFs reduces model performance due to attention dispersion and structure loss. This paper proposes a structure-preserving method that encodes document elements using the LATEX paradigm. Through experiments on four benchmark datasets and four mainstream models, this method significantly improves the model's accuracy in document question-answering tasks.

**Strengths:**

1. By using the latex paradigm to encode document structure, it only adjusts the input format to take effect, without modifying the model architecture or conducting additional training, resulting in low implementation costs.

2. Four benchmark datasets covering multiple scenarios such as MMLongBench are selected, and four mainstream MLLMs are used for testing, ensuring high reliability of the results.

**Weaknesses:**

1. The method relies on the LATEX paradigm, which may have limitations in encoding complex document structures, such as irregular layouts, handwritten annotations.

2. The ablation experiments show that structured text alone performs poorly; the method must rely on image input to be effective, which limits its application in scenarios where only text data is available.

3. The text extraction process uses OCR for image-based PDFs, but this paper does not discuss how OCR errors affect the quality of structured text.

4. All experiments are conducted on 7B-scale models, but there is no exploration of whether the method's effectiveness varies with model size.

**Questions:**

see weaknesses

---

### Meta-Review · Area_Chair_uGcz · 2025-12-23

**Summary:**

The paper investigates how input format, specifically using LaTeX-structured text alongside images, enhances multimodal large language models' (MLLMs) document question-answering performance by reducing attention dispersion and preserving layout. Reviewers generally acknowledge the paper's clear presentation, comprehensive experiments across four benchmarks (e.g., MMLongBench) and multiple 7B-scale models, and the practical advantage of being training-free. However, significant concerns are raised about novelty, as the core observation—plain text harming performance due to structure loss—is considered unsurprising and incremental, with the LaTeX approach seen as a simple prompt engineering trick rather than a technical innovation. Methodological issues include underspecified details (e.g., the model used for generating structured text), lack of efficiency analysis (e.g., token counts or latency), and limited generalization due to testing only on 7B models without larger or proprietary models. Reviewers also note weak baselines (e.g., no comparison to HTML/XML formats) and overinterpreted attention analysis, which lacks rigorous causal evidence. While one reviewer rated the paper marginally below acceptance (4/10), three others recommended rejection (2/10), citing poor contribution and soundness. Overall, the paper offers a useful empirical demonstration but falls short of ICLR's standards due to minimal novelty and unresolved methodological gaps.

**Reviewer Concerns:**

No formal rebuttal is provided.

**Reviewer Scores:**

No formal rebuttal is provided.

---

### Decision · Program_Chairs · 2026-01-26

Reject